# Can We Edit Factual Knowledge by In-Context Learning?

**Ce Zheng[1], Lei Li[1], Qingxiu Dong[1], Yuxuan Fan[1],**
**Zhiyong Wu[2], Jingjing Xu[2] and Baobao Chang[1*]**
[1] National Key Laboratory for Multimedia Information Processing,
School of Computer Science, Peking University
[2] Shanghai Artificial Intelligence Laboratory
{zce1112zslx,jingjingxu,chbb}@pku.edu.cn, nlp.lilei@gmail.com
{dqx,yxfan}@stu.pku.edu.cn, wuzhiyong@pjlab.org.cn

## Abstract

Previous studies have shown that large language models (LLMs) like GPTs store massive factual knowledge in their parameters. However, the stored knowledge could be false or outdated. Traditional knowledge editing methods refine LLMs via fine-tuning on texts containing specific knowledge. However, with the increasing scales of LLMs, these gradient-based approaches bring large computation costs. The trend of model-as-a-service also makes it impossible to modify knowledge in black-box LLMs. Inspired by in-context learning (ICL), a new paradigm based on demonstration contexts without parameter updating, we explore whether ICL can edit factual knowledge. To answer this question, we give a comprehensive empirical study of ICL strategies. Experiments show that in-context knowledge editing (IKE), without any gradient and parameter updating, achieves a competitive success rate compared to gradient-based methods on GPT-J (6B) but with much fewer side effects, including less over-editing on similar but unrelated facts and less knowledge forgetting on previously stored knowledge. We also apply the method to larger LMs with tens or hundreds of parameters like OPT-175B, which shows the scalability of our method. The code is available at https://github.com/pkunlp-icler/IKE.

## 1 Introduction

Pre-trained Language models (LMs) have set a new paradigm for NLP research and sweep across all existing NLP benchmarks. Due to their promising results, researchers have empowered LMs with new skills that meet real-world needs, such as using web browsers (Nakano et al., 2021), coding (Chen et al., 2021), playing strategic game (FAIR et al., 2022), and conversational talents (OpenAI, 2022, 2023). However, the wide application of LMs also raises growing concerns regarding its pitfall of generating content that is fake (Elazar et al., 2021; Cao

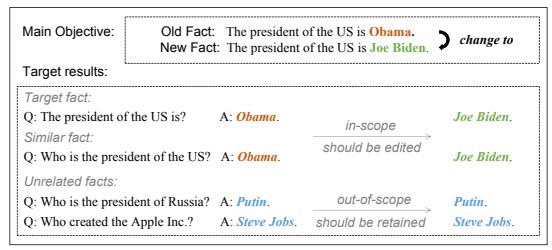

Figure 1: An illustration of knowledge editing, which requires generalization to different prompts describing the same fact without interference on other facts.

et al., 2021a), out-dated (Dhingra et al., 2022), biased (Sheng et al., 2019; Zhao et al., 2021), and offensive (Gehman et al., 2020). To mitigate this pitfall, knowledge editing (Fig. 1) aiming to modify the knowledge learned of LMs has attracted increasing attention (Mitchell et al., 2022a; Meng et al., 2022a). The goal of knowledge editing is two-fold: *generalization* and *specificity*. The former requires generalizing to various prompts describing the same knowledge and the latter requires no interference with other unrelated knowledge.

Previous knowledge editing methods mainly adopt gradient-based methods to modify specific model parameters for a desired model behavior (Mitchell et al., 2022a; Meng et al., 2022a), e.g., updating the president after the election. However, the identification of the target knowledge neurons usually requires gradient estimation with heavy computation overhead (Dai et al., 2022). In addition, the updated parameters inherently lead to side effects beyond the desired editions, such as forgetting previously-learned facts or over-editing on unrelated facts. Previous studies have shown that when a large-scale LM (LLM) is deployed as a black-box service (Sun et al., 2022), a minor modification to its parameters could dramatically influence its behavior for end users. Therefore, traditional methods still suffer from editing LLMs

---
*Corresponding author

since these limitations impede the scalability and generalizability.

Recently, in-context learning (ICL) (Brown et al., 2020) has emerged as a new paradigm for instructing LLMs to perform complex tasks. In ICL, the task description and demonstration examples are represented in natural language to form a context, and the prediction of LMs conditioned on the context is transformed into answers according to predefined rules (Brown et al., 2020). In this way, large LMs adapt to various downstream tasks without any modifications to parameters, making it a natural fit for knowledge editing on large LMs. First, it reduces the computation overhead by avoiding modifications to parameters, as well as eliminates the risk of side effects introduced by parameter updates. Most importantly, ICL provides an interpretable way for humans to calibrate LM behaviors. Despite these advantages, whether ICL is applicable to knowledge editing still remains unclear.

In this paper, we investigate the potential of ICL to perform knowledge editing for LLMs. We focus on two goals: (1) ensuring generalization, so that large LMs can generalize to various text surfaces for a piece of updated knowledge, and (2) ensuring specificity, by making accurate modifications to the target knowledge fact while preserving other irrelevant facts. To achieve these goals simultaneously, we design demonstration formatting and organization strategies to construct suitable in-context learning demonstrations for guiding knowledge editing on LLMs. We define three types of demonstration formatting templates including (i) *copy*, which aims to inject new facts into LMs; (ii) *update*, which improves the generalization of injected knowledge fact; and (iii) *retain*, which guides LMs to preserve unrelated knowledge facts. Additionally, to fully harness the potential of ICL for knowledge editing, we retrieve relevant knowledge facts from the training corpus as demonstration inputs.

Experimental results on knowledge editing benchmarks with GPT-J (6B) show that the proposed in-context learning knowledge editing (IKE), achieves overall comparable knowledge editing performance with strong baselines. For example, IKE outperforms MEND (Mitchell et al., 2022a) by an absolute 10% editing success rate and obtains 30 points gain regarding the specificity over ROME (Meng et al., 2022a). As there are no parameter modifications, IKE is applicable to LLMs

such as OPT-175B and exhibits better memorization ability, i.e., after editing, nearly 50% knowledge facts retain relatively high probability. Further analysis reveals that demonstration selection and the *retain* demonstrations contribute to specificity, while the *update* demonstrations improve generalization. Finally, we discuss the potential challenges that IKE may encounter when applied in real-world scenarios, and provide corresponding discussions.

In summary, the contributions of this study are four-fold:

- To the best of our knowledge, this work represents the first systematic exploration of the potential for ICL to edit knowledge in LMs.

- We give comprehensive empirical studies on ICL strategies and analyze how these strategies affect the final performance.

- By designing proper demonstration formatting and organization strategies, IKE achieves comparable success rates with less computation overhead and side effects.

- We investigate the feasibility of applying IKE to real-world scenarios and discuss potential challenges.

## 2 Related Work

**Knowledge Editing Methods**  Recent studies on knowledge editing are mostly hype-network-based or attribution-based (Yao et al., 2023). The hype-network-based methods train a hyper-network to get gradient changes for certain edits. For example, Cao et al. (2021b) used a hyper-network to predict parameter shift at test time, which alters a fact while retaining unrelated facts. MEND (Mitchell et al., 2022a) learned to transform the original fine-tuning gradient into a low-rank decomposition of the gradient. Mitchell et al. (2022b) used an edit memory retriever and a counterfactual model to generate without updating the parameters of the base model. Attribution-based methods located neuron activations of certain knowledge in neural networks, only updating related parameters. Dai et al. (2022) evaluated the contribution of different neurons to specific knowledge using gradient-based attributions, and updated or erased facts by replacing columns in Multilayer Perceptron(MLP) weight matrices with scaled embedding vectors. Meng et al. (2022a) located single layer that expresses factual knowledge, and edited such factual

knowledge by writing new key-value pair in MLP module.

**Knowledge Editing Benchmarks** Several knowledge editing benchmarks are commonly used to evaluate the efficacy and specificity of editing approaches. For BERT-style models, fact-checking dataset FEVER (Thorne et al., 2018) and question-answer dataset zsRE (Levy et al., 2017) are usually adopted. In FEVER, each $x$ is a claim and each $y$ indicates the validity of corresponding claim. In zsRE, each $x$ is a question about a fact and each $y$ is the answer, and $x_{loc}$ questions fact irrelevant to $x$. For GPT-style models, Mitchell et al. (2022a) introduced Wikitext editing dataset that requests the model to complete passage with edited continuation while the distribution of each token is unrelated passage $x_{loc}$ should remain unchanged. In our experiment, we use a more challenging QA dataset called COUNTERFACT (Meng et al., 2022a). In COUNTERFACT, the edited answer $y$ to question $x$ can sometimes be counterfactual to real world, and unrelated out-of-scope sample $x_{loc}$ is much more difficult than that in zsRE, which makes it harder for the model to predict desired answer. Furthermore, these desired facts are hardly captured by pre-trained LMs, avoiding the effects of LLMs knowing this knowledge before editing.

**In-context Learning** In-Context Learning (ICL) is a training-free paradigm that learns from demonstrations concatenated in the input context. Given related examples and a query, the model learns from analogy to make predictions (Brown et al., 2020; Liu et al., 2022). Existing knowledge editing methods require re-calculating the gradient or calculating and perform such knowledge editing in an inexpensive way. Si et al. (2022) is the first to explore whether in-context learning can update knowledge in LLMs, and show that incorporating all kinds of demonstration increase the success rate of knowledge editing. However, they only focus on GPT-3, without deep exploration on the potential ability and side effects of knowledge editing.

## 3 Task Formulation

The goal of knowledge editing is to inject a new fact $(x^*, y^*)$ into a LM $\mathcal{M}$ by maximizing the probability $\mathcal{P}_{\mathcal{M}}(y^*|x^*)$. The $x^*$ is the prompt to probe the factual knowledge in $\mathcal{M}$ (e.g., The president of the US is), and $y^*$ will be the editing target Joe Biden. Knowledge editing also requires

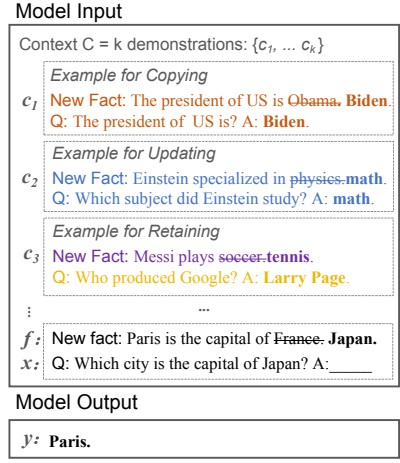

Figure 2: An illustration of in-context knowledge editing.

generalization and specificity:

- **Generalization**: For the prompt $x$ in the edit scope $\mathcal{D}_{x^*}$ (i.e., prompts related to the new fact), the prediction of $x \in \mathcal{D}_{x^*}$ should be also updated to $y^*$. For example, the prediction of Q: Who is the president of the US? A: will be updated to Joe Biden.

- **Specificity**: For the prompt $x$ out of the edit scope, $x \notin \mathcal{D}_{x^*}$, the prediction of $x$ should be its original prediction $y^o$. For example, the prediction of The president of Russia is should be retained.

## 4 Method: IKE

### 4.1 In-Context Learning

In-Context Learning (ICL) is proposed by Brown et al. (2020) for few-shot learning. For a large language model $\mathcal{M}$, ICL aims to predict $\hat{y} \in \mathcal{Y}$ for an input $x$ without any parameter updating based on $k$ demonstrations $C = \{(x_1, y_1), \ldots, (x_k, y_k)\}$. The language model $\mathcal{M}$ predicts the probability of $y \in \mathcal{Y}$ given $x$: $\mathcal{P}_{\mathcal{M}}(y \mid x, C)$. More specifically, ICL uses templates $\mathcal{T}$ to transform the inputs and labels into natural language texts. Take sentiment analysis as an example, an in-context demonstration with input $x_i$ and label $y_i$ will be transformed to **Sentence:** $x_i$**. Sentiment:** $y_i$, then the language model $\mathcal{M}$ will predict $y \in \mathcal{Y}$ given $\mathcal{T}(x_1, y_1), \ldots, \mathcal{T}(x_k, y_k), \mathcal{T}(x, )$.

### 4.2 In-Context Knowledge Editing

When we inject a target fact $f = (x^*, y^*)$ into LMs, we will construct $k$ demonstrations $C =$

$\{c_1, \ldots, c_k\}$. The goal of knowledge editing is to maximize $\mathcal{P}(y^* \mid x, f, C)$ when prompt $x$ is in the editing scope of target prompt $x^*$, $x \in \mathcal{D}_{x^*}$ (the Generalization goal) and minimize the distance between $\mathcal{P}(y \mid x, f, C)$ and $\mathcal{P}(y \mid x)$ when $x \notin \mathcal{D}_{x^*}$ (the Specificity goal). LMs should determine whether the probing prompt $x$ is in the editing scope of $x^*$, namely $\mathcal{D}_{x^*}$. To achieve these goals with ICL, proper demonstration inputs are crucial. We further decompose the demonstration construction for knowledge editing with $f$ as the target into two sub-problems:

(i) how to design the format of each demonstration; and (ii) how to select and rank in-context demonstrations (Dong et al., 2023).

### 4.2.1 Demonstration Formatting

Each demonstration $c_i$ contains a new fact $f_i = (x_i^*, y_i^*)$, a probing prompt $x_i$ and its prediction $y_i$. In-context demonstrations should teach LMs to *copy, update* and *retain* the predictions for different prompts:

- *copy*: To inject new facts into LMs, the first step is to teach them to copy the prediction of the target prompt in new facts. In *copy* demonstrations, $x_i = x_i^*$ and $y_i = y_i^*$.

- *update*: Knowledge editing is not simply teaching LMs to repeat the new fact. For the generalization of knowledge editing, the prediction of prompts in the editing scope should also be updated. In *update* demonstrations, $x_i \in \mathcal{D}_{x_i^*}$ and $y_i = y_i^*$.

- *retain*: For the specificity of knowledge editing, LMs should keep their original prediction in out-of-scope prompts. In *retain* demonstrations, $x_i \notin \mathcal{D}_{x_i^*}$ and $y_i$ should be its original answer $y_i^o$.

The template $\mathcal{T}$ of IKE transforms $f$, $x$ and $y$ into natural language: $\mathcal{T}(f, x, y) =$ New Fact: $f$. Prompt: $x$ $y$. Details are listed in §A.

### 4.2.2 Demonstration Organization

When we edit a knowledge fact $f$ in LMs, we construct $k$ demonstrations $C = \{c_1, \ldots, c_k\}$ from the training corpus. Which demonstrations are good demonstrations for in-context editing? We follow Liu et al. (2022) to use an unsupervised retriever to choose $k$ nearest neighbors. More specifically, we use a pretrained sentence encoder $\mathcal{E}$ to encode

| Editing Method | Scalability | Side Effects | Interpretability |
|---|---|---|---|
| Gradient-based | + | - - - | + |
| In-context Learning | +++ | - | +++ |

Table 1: Comparison of knowledge editing methods, ICL is more computationally efficient and interpretable, with fewer side effects introduced.

the prompt $x^*$ of new fact $f$ together with its original answer $y^o$ and targeted prediction $y^*$. The records in the training corpus will be encoded in the same way and $k$-NN facts are retrieved based on the cosine similarity. The ranking of in-context demonstrations also depends on the cosine similarity: $\cos(c_0, f) < \cos(c_1, f) < \ldots < \cos(c_k, f)$, where $c_1, \ldots, c_k$ are placed in the context from left to right.

### 4.3 Discussion: Gradient-based methods and gradient-free methods

Previous parameter updating methods will adjust the parameters $\theta$ of LMs $\mathcal{M}$. They calculate $\Delta\theta$ based on the gradients $\nabla_\theta - \log \mathcal{P}_\mathcal{M}(y^* \mid x^*)$ to update the base model $\mathcal{M}_\theta$ to a edited one $\mathcal{M}'_{\theta + \Delta\theta}$. The editing method will then be evaluated by $\mathcal{P}_{\mathcal{M}'}(y \mid x)$. Instead, in-context learning modifies the knowledge fact in $\mathcal{M}$ by constructing demonstrations $C$ for the new fact $f = (x^*, y^*)$, then the editing method will be evaluated by $\mathcal{P}_\mathcal{M}(y \mid x, f, C)$. Comparing $\mathcal{P}_\mathcal{M}(y \mid x, f, C)$ with $\mathcal{P}_{\mathcal{M}'}(y \mid x)$, it can be found that: (i) ICL requires no gradient estimation for the target fact and keeps the original LM $\mathcal{M}$ untouched after knowledge editing. This greatly reduces the computation overhead thus making the editing applicable for LMs with trillion-level parameters, as well as eliminating the side effects of the modified parameters. (ii) The demonstration $C$ is represented in the natural text which is more interpretable than the salient parameter update $\Delta\theta$. It provides a human-understandable interface for calibrating the model behavior. We highlight the characteristics of these two methods in Table 1.

## 5 Experiment

In this section, we perform experiments to answer the following research question:

- Compared to gradient-based methods, what's the performance of IKE?

- How do the demonstration designing strategies influence the performance of IKE?

- How does the scale of LMs affect the performance of IKE, can IKE scale up to large language models with tens or hundreds of billions of parameters?

- What are the side effects of knowledge editing and does IKE cause more or fewer side effects than other parameter updating methods?

We first introduce the experimental settings including the compared baseline methods, evaluation benchmark, and LMs across different scales for knowledge editing (§5.1). We then analyze the main knowledge editing results in §5.2 and the impacting factors of in-context learning knowledge editing (§5.3).

## 5.1 Experimental Setting

We aim to evaluate the performance of in-context knowledge editing compared to parameter updating approaches. We also conduct experiments on different sizes of LMs to explore the scaling-up ability of in-context knowledge editing.

### 5.1.1 Baselines

Following previous knowledge-editing methods, we also choose GPT-J (6B) as our main evaluation backbone. The compared baselines include:

**FT** Fine-tuning the base model on text describing the edit fact, without training a new model editor by applying Adam with early stopping.

**MEND** MEND (Mitchell et al., 2022a) transforms the fine-tuning gradient of an updated fact by decomposing the weight matrix into rank-1 form with the pretrained hyper-network.

**ROME** ROME (Meng et al., 2022a) learns to locate factual retrievals of a specific set of MLP modules and update knowledge by directly writing in new key-value pairs in the MLP module.

**PROMPT** To explore how in-context demonstrations influence the performance of IKE. We directly use the new fact as context to probe the LMs by $\mathcal{P}(y|x, f)$ where $f = (x^*, y^*)$.

The implementation details are in §A

### 5.1.2 Evaluation Setup

**Models** To explore how the scale of LMs will influence the effectiveness of in-context knowledge editing, we evaluate in-context knowledge editing on five GPT-like auto-regressive transformer language models whose scales range from 1.5B to 175B parameters:

- GPT-2 XL (1.5B) (Radford et al., 2019), the 1.5 billion parameter version of GPT-2.

- GPT-NEO (2.7B) (Gao et al., 2021), the 2.7 billion parameter version of a GPT-2 like causal language model released by EleutherAI. It is trained on the Pile dataset specifically designed for LLM training.

- GPT-J (6B) (Wang and Komatsuzaki, 2021), an auto-regressive text generation model trained on the Pile with 6 billion parameters.

- GPT-NEOX (20B) (Black et al., 2022), a 20 billion parameter auto-regressive language model trained on the Pile.

- OPT (175B) (Zhang et al., 2022), open pretrained transformers with 175 billion parameters created by MetaAI.

**Benchmark** We mainly evaluate baselines on COUNTERFACT (Meng et al., 2022a), a challenging benchmark suitable for GPT-like causal language models with difficult editing targets and hard-to-distinguish editing scopes. It contains $21,919$ records of diverse relations and entities. The goal of each record is to change the knowledge triplet $(s^*, r^*, o^c)$ to $(s^*, r^*, o^*)$ where $s^*$ and $r^*$ are described by the target prompt $x^*$. The record also contains paraphrase prompts $P^P$ as in-scope prompts and neighborhood prompts $P^N$, i.e., knowledge triplets $(s', r^*, o^c)$ that share the same object with target triplets as out-of-scope prompts. We follow Meng et al. (2022a) to use first 2000 records as the test set and the remaining records are divided into training set. The details of COUNTERFACT are listed in §B.

**Metrics** The performance of knowledge editing is measured from three aspects (*Efficacy*, *Generalization*, and *Specificity*).

- *Efficacy* measures the post-editing accuracy for target prompts by Efficacy Score (**ES**, $\mathbb{E}[\mathbb{I}[\mathcal{P}(o^*) > \mathcal{P}(o^c)]]$) and Efficacy Magnitude (**EM**, $\mathbb{E}[\mathcal{P}(o^*) - \mathcal{P}(o^c)]$).

- *Generalization* measures post-editing accuracy on paraphrase prompts by Paraphrase

| Editing Method | #Edited Params. | #Extra Params. | Score S↑ | Efficacy | | Generalization | | Specificity | |
|---|---|---|---|---|---|---|---|---|---|
| | | | | ES↑ | EM↑ | PS↑ | PM↑ | NS↑ | NM↑ |
| GPT-J (6B) | 0 | 0 | 22.0 | 16.2 | -7.4 | 15.9 | -7.5 | 83.2 | 7.4 |
| FT | 64M | 0 | 28.7 | 99.9 | 98.6 | 96.4 | 67.0 | **11.9** | **-48.6** |
| MEND | 384M | 896M | 63.6 | 90.4 | 53.9 | **53.4** | **14.3** | **57.6** | **-3.3** |
| ROME | 64M | 256M | **91.5** | **100** | **99.4** | **99.6** | **78.0** | 78.5 | 5.0 |
| PROMPT | 0 | 0 | 63.3 | 99.7 | 80.9 | 91.0 | 32.9 | **37.9** | **-2.8** |
| IKE (32 examples) | 0 | 20M | 89.6 | **100** | 91.7 | 95.2 | 64.5 | 77.0 | **35.2** |
| OPT (175B) | 0 | 0 | 18.7 | 12.6 | -8.4 | 14.3 | -8.1 | 86.9 | 8.4 |
| PROMPT | 0 | 0 | 58.1 | 99.6 | 77.2 | 94.1 | 37.4 | **32.3** | **-7.8** |
| IKE (32 examples) | 0 | 20M | **94.1** | **100** | **92.5** | **98.8** | **83.6** | **85.1** | **45.5** |

Table 2: Knowledge Editing Performance for GPT-J (6B) and OPT (175B) on COUNTERFACT. Efficacy, Generalization, and Specificity are evaluated based on target, in-scope, and out-of-scope prompts respectively. Details of the Metric can be found in §5.1.2. **green** means column-wise maxima and **red** indicates poor generalization or specificity.

Score (**PS**) and Paraphrase Magnitude (**PM**). The definition of PS and PM is similar to ES and EM.

- *Specificity* measures the accuracy of neighborhood prompts by Neighborhood Score (**NS**, $\mathbb{E}[\mathbb{I}[\mathcal{P}(o^c) > \mathcal{P}(o^*)]]$) and Neighborhood Magnitude (**NM**, $\mathbb{E}[\mathcal{P}(o^c) - \mathcal{P}(o^*)]$), as the neighborhood prompts $(s', r^*, o^c)$ share the same original object with the target prompt and these facts are not supposed to be edited.

We also follow Meng et al. (2022a) to report the harmonic mean of ES, PS, NS as Score (**S**)

## 5.2 Main Results

The top rows of Table 2 show the knowledge editing results of different methods. Our findings are: (i) All methods perform well in terms of efficacy, as indicated by their close ES scores. However, there are significant differences in terms of generalization and specificity. For instance, **FT** achieves high ES (99.9) and PS (96.4) scores but performs poorly in terms of specificity. This highlights the challenge of balancing generalization and specificity in knowledge editing. (ii) Among the baseline methods, **ROME** performs the best overall regarding all three metrics, but comes with high computational overheads. Due to this limitation, it is not applicable to larger LMs such as OPT-175B that are in more urgent need of knowledge editing. (iii) The proposed method **IKE** excels in specificity but also performs well in efficacy and generalization. For example, IKE achieves a comparable overall score with ROME on GPT-J (89.6 v.s. 91.5), while requiring no parameter

| Editing Method | S↑ | ES↑ | PS↑ | NS↑ |
|---|---|---|---|---|
| IKE (32 examples) | **89.6** | **100** | 95.2 | 77.0 |
| - 4 examples | 81.5 | 99.6 | 83.5 | 67.5 |
| - 8 examples | 84.2 | 100 | 85.6 | 71.7 |
| - 16 examples | 87.0 | 100 | 91.7 | 73.6 |
| - random selection | **70.3** | 100 | 95.8 | **45.0** |
| - random ordering | 88.9 | 100 | 95.4 | 75.1 |
| - *w/o copy* | 88.6 | 100 | 96.9 | 73.9 |
| - *w/o update* | 84.4 | 100 | **73.8** | **83.4** |
| - *w/o retain* | **28.0** | 100 | **99.8** | **11.5** |

Table 3: Ablation study on demonstration designing. Increasing the number of demonstrations improves the overall performance. The definitions of metrics are the same as Table 2. Demonstration selection and the *retain* demonstrations contribute to specificity, while the *update* demonstrations improve generalization.

modifications on LMs. This computation benefit makes it possible to perform knowledge editing on large LMs such as OPT-175B, where **IKE** achieves clear improvements over **PROMPT** by 36.0 points. These results demonstrate the effectiveness, efficiency and scalability of IKE in knowledge editing.

## 5.3 Analysis

In this part, we discuss the effects of different demonstration strategies, the scalability of IKE for models across scales and side effects introduced by knowledge editing.

### 5.3.1 Ablation on Demonstration

**Demonstration Numbers** The number of demonstrations is one of the influencing factors for the ICL performance (Brown et al., 2020). We investigate how the number of demonstrations influences the IKE performance in the second

| Models | Generalization | | Specificity | |
|---|---|---|---|---|
| | PS↑ | PM↑ | NS↑ | NM↑ |
| GPT-2 XL (1.5B) | 85.1 | 42.8 | 72.0 | 21.0 |
| GPT-NEO (2.7B) | 96.3 | 73.5 | 70.7 | 28.0 |
| GPT-J (6B) | 95.2 | 64.5 | 77.0 | 35.2 |
| GPT-NEOX (20B) | 97.5 | 78.3 | 79.8 | 41.3 |
| OPT (175B) | 98.8 | 83.6 | 85.1 | 45.5 |

Table 4: The IKE performance on different LMs whose scales range from 1.5B to 175B. Larger LMs achieve better generalization and specificity. Detailed discussion can be found in §C.

block in Table 3. Without any demonstrations, **PROMPT** exhibits over-generalization for its low NS (37.9), indicating it simply learns to copy the prediction. Given a few demonstrations (4 or 8), **IKE** performs worse than PROMPT in Efficacy and Generalization as it begins to distinguish whether a prompt is in the editing scope. With the increased number of demonstrations, IKE gradually learns to balance generalization and specificity, achieving a better trade-off.

**Demonstration Organization** Previous studies (Liu et al., 2022; Rubin et al., 2022; Lu et al., 2022) suggest that demonstration organization including Demonstration Selection and Demonstration Ordering (Dong et al., 2023) is also crucial for ICL. Our proposal follows a simple unsupervised method Liu et al. (2022), to retrieve and order demonstrations from the training corpus based on the cosine similarities between the input prompt and demonstrations. In our two ablation studies in the third block of Table 3, we find that removing the selection procedure (i.e., *Random Selection*) leads to a clear drop in the NS score from 77.0 to 45.0, indicating the importance of proper prompt selection. However, *random ordering* brings negligible performance difference. We speculate that this is because the selected prompts are highly related to the target fact and the attention mechanism in Transformer-based LMs can handle long-range dependencies well. We leave further improvements as future work.

**Demonstration Formatting** We further examine the impact of demonstration types including *copy*, *update* and *retain*. As shown in the fourth block in Table 3, removing *copy* demonstrations causes slight performance degradation, as LMs can easily copy the content in the demonstration even without a *copy* demonstration. Instead, *update* demonstrations perform an important role in teaching LMs

to modify their knowledge, as indicated by a much poorer generalization score after removing *upate* demonstrations. Besides, The removal of *retain* demonstrations leads to a dramatic drop in the specificity, as measured by the NM score, which decreases from 35.2 to -47.6. This indicates that *retain* demonstrations are crucial in helping LMs identify out-of-scope facts and maintain their original predictions on those prompts.

### 5.3.2 IKE Benefits from Model Scaling

We further evaluate IKE on COUNTERFACT for five GPT-like causal language models across different scales. As previous experiments have shown that all methods exhibit high knowledge editing efficacy, we focus on the generalization and specificity for large LMs, as these metrics are defined to measure the side effects that could cause great influences on end users. As demonstrated in Table 4, we find that the performance of IKE is positively correlated with the scale of the LM and the largest OPT-175B achieves the strongest generalization and specificity results. This is inspiring as the performance IKE could be enhanced with the increased scale of LMs, making it pluggable for future stronger LM backbones.

### 5.3.3 Resilience to Over-Editing

Over-editing is a common side effect of knowledge editing, which denotes the influences on out-of-scope facts when editing a targeted fact. Although COUNTERFACT already includes out-of-scope prompts consisting of $(s', r^*, o^c)$ sharing the same relation $r$ and original object $o^c$ with the editing target: $(s^*, r^*, o^c) \rightarrow (s^*, r^*, o^*)$, we perform a more comprehensive evaluation on over-editing by adopting the **contrastive knowledge assessment** (CKA) proposed by Dong et al. (2022). Specifically, for a triplet $(s, r, o)$, CKA replaces $r$ with other similar but unrelated relations $r'$ and compares $\mathcal{P}_{\mathcal{M}}(o|s, r)$ and $\mathcal{P}_{\mathcal{M}}(o|s, r')$ to assess whether $\mathcal{M}$ knows the fact $(s, r, o)$. Inspired by this, we regard $(s^*, r', o^*)$ as similar but unrelated prompts and consider the change in $\mathcal{P}(o^*|s^*, r')$ and find that $\mathcal{P}(o^*|s^*, r')$ will also increase after injecting $(s^*, r^*, o^*)$. To further explore over-editing in different methods, we consider the CKA score, $\mathcal{P}(o^*|s^*, r^*)/\mathbb{E}_{r' \in \mathcal{R}} \mathcal{P}(o^*|s^*, r')$.

The results of CKA evaluation are listed in Table 5. If the CKA score is less than predefined threshold $\alpha$, the perplexity of the correct fact is close to the perplexity of contrastive fake facts,

| Method | CKA Score (↑) | False Rate (score $< \alpha$) (↓) | |
| --- | --- | --- | --- |
| | | $\alpha$ =1.0 | $\alpha$ =1.1 |
| FT | 1.8 | 0.6 % | 19.5 % |
| ROME | **1.7** | 0.4 % | **24.1 %** |
| PROMPT | **2.3** | 0.2 % | **1.0 %** |
| IKE | 2.1 | **0.1 %** | 1.7 % |

Table 5: CKA Evaluation shows that editing methods will over-edit $(s^*, r', *)$ when editing $(s^*, r, o) \rightarrow (s^*, r, o^*)$. Low CKA score means over-generalization and False Rate is the fraction of records whose score is less than $\alpha$.

| Method | Prob. Drop (↓) | Forgetting Rate (↓) |
| --- | --- | --- |
| FT | 7.6 | 94.1 % |
| ROME | 7.7 | **99.3 %** |
| PROMPT | 6.2 | 64.1 % |
| IKE | **6.1** | **50.5 %** |

Table 6: Knowledge Editing can cause forgetting of original facts in LMs. Prob. Drop means $\Delta \mathcal{P}(o^c|s^*, r)$ between pre- and post-editing. An original fact is forgotten when $\Delta \mathcal{P}(o^c|s^*, r^*) > 0.5 \times \mathcal{P}(o^c|s^*, r^*)$.

which turns out to be an editing failure. Although all baselines perform well in terms of editing efficacy, they tend to be over-generalization under a stricter contrastive assessment. ROME gets the lowest average CKA score and highest false rate, which shows its poor ability to identify out-of-scope prompts sharing the same subject with target prompts. IKE has less influence on over-editing.

### 5.3.4 Maintenance for Original Knowledge

We conclude that previous factual knowledge stored in LMs will be erased or forgotten in knowledge editing. We consider the change of $\mathcal{P}(o^c|s^*, r)$ before and after editing in Table 6. The results demonstrate that all editing methods will cause the drop of $\mathcal{P}(o^c|s^*, r^*)$. ROME forgets almost all original facts. If we want to correct the prediction of LMs, erasing the original factual knowledge is necessary. However, if we want to update the prediction of language models like updating the prediction of `The president of US is` from `Donald Trump` to `Joe Biden` (time-aware relations), the old knowledge `In 2017, the president of US was Donald Trump` should not be forgotten.

To evaluate the forgetting of such time-aware knowledge in editing, we construct a small benchmark based on TEMPLAMA (Dhingra et al., 2022) to further show that IKE can cause less knowledge forgetting than other baselines in §D.

## 6 Discussions

In previous experiments, we follow the setup of previous studies Meng et al. (2022a) and mainly evaluate methods to edit individual facts for a fair comparison. Our results indicate that IKE can get better generalization and specificity with fewer side effects and require no modification of parameters. Nevertheless, in order to investigate the feasibility

of applying IKE to real-world scenarios, several important questions remain under-explored: (1) **Can IKE be extended to accommodate a larger number of editing facts?** Considering the limited input length of language models, it may not be feasible to include tremendous editing facts within the context. (2) **Can IKE be adapted to handle different formats and domains of facts and prompts?** In IKE, the domain and format of facts and prompts are kept consistent. However, in real-world settings, facts and prompts come in diverse forms.

Mitchell et al. (2022b) propose a retrieval-based method for editing multiple knowledge facts. Similarly, IKE with an external memory to store factual edits can retrieve the proper factual edit to construct context for a given prompt, thus avoid prepending all factual edits in context forever. To validate the generalization of IKE on different forms of facts or prompts, we replaced facts with neutral data from Wikipedia, or replaced prompts with generation prompts that prompt the LM to generate text related to the new object. Detailed discussion can be found in §E.

## 7 Conclusion

In this work, we examine the potential of in-context learning for knowledge editing on large-scale language models. Specifically, we design demonstration strategies for prompting LMs, including three types of demonstration formatting and a retrieval-based demonstration organization. We show that the proposed method, IKE, achieves competitive knowledge editing efficacy without requiring any parameter modifications, as well as maintains decent generalization and specificity performance. Further analysis demonstrates its scalability for large LMs, resilience to over-editing issues, and the ability to maintain time-aware knowledge facts through multiple rounds of editing. Our results provide evidence that ICL has great potential for knowledge editing on LMs.

## Limitations

The limitations of our work are summarized as follow: The limitations of our work primarily include the following aspects:

1. We only discuss modifications on factual knowledge in language models and do not address the editing of other types of knowledge like commonsense knowledge.

2. Introducing ICL in our method brings additional computational costs due to the longer context used.

3. Despite discussing some of the side effects of knowledge editing, the mechanism of ICL is still unclear, and there may be potential risks associated with modifying the knowledge in the model.

4. Although we provide some preliminary discussions on applying IKE in practical scenarios, there is still a considerable way to go before real-world applications can be realized.

## Acknowledgements

This paper is supported by the National Key R&D Program of China under Grand No.2018AAA0102003, and the National Science Foundation of China under Grant No.61936012.

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

## A Implementation Details

### A.1 IKE

We implement IKE with PyTorch (Paszke et al., 2017), Huggingface transformers (Wolf et al., 2019), and sentence transformers (Reimers and Gurevych, 2019). Pytorch is licensed under the modified BSD license. Huggingface and Sentence transformers are under Apache License 2.0. IKE with 32 examples are run in a 40 GB NVIDIA A40 GPU for about 3 GPU hours.

### A.2 Demonstration Designing

We follow Liu et al. (2022) to choose $k$-NN examples from the training corpus. The demonstrations are encoded by `all-MiniLM-L6-v2`. For LMs with maximum context length as 2048, we set $k$ to 32; and for LMs with maximum context length as 1024, we set $k$ to 16.

#### A.2.1 Demonstration Formatting

We have defined three types of in-context demonstrations in 4.2.1. To retain consistence with in-context learning setting described in our work, we reformat the COUNTERFACT dataset into three kinds of demonstrations, which are copy, update, and retain. Examples are shown in table 7. Here the true fact to be changed is "What does Sylvano Bussotti play? They play opera.", the new fact is "What does Sylvano Bussotti play? They play jazz.". The demonstration format follows $\mathcal{T}(f, x, y) =$ New Fact: $f$. Prompt: $x\ y$, where $f$ is the new fact, $x$ is the probing prompt (e.g. What does Sylvano Bussotti play? They play) and $y$ is model prediction (e.g. jazz). Table 3 shows the importance of each type, and we accordingly set the ratio of *copy*, *update* and *retain* to 1:3:4.

The order of demonstration types is an underexplored influencing factor of IKE. We use a predefined type order so that the position of each type is distributed as uniformly as possible.

### A.3 Other Baselines

We conduct other baselines with the code implemented by Meng et al. (2022a). [1] We simply add the prefix `Prompt:` in prompts and report the results conducted by us.

## B Details of COUNTERFACT Dataset

Table 8 illustrates an example from COUNTERFACT. This entry requests that "the mother tongue of Danielle Darrieux should be changed from English to French". Each entry has several paraphrase prompts and several neighborhood prompts. Paraphrase prompts are semantically equivalent to the original prompt, neighborhood prompts are those that share the same relation and object with the original prompt but have different subjects. The raw COUNTERFACT dataset also includes attribute prompts and generation prompts, but they are not adopted in our work. We use the first 2,000 records as test split for evaluation and other records are training split.

## C Model Details

In Table 4, we discussed the performance of IKE on five different decoder-only GPT-like models at various scales. All models use 32 samples except for GPT-2 XL, which is limited by its maximum input length. The existing results indicate that on these models that have emerging ICL capabilities, IKE benefits from the increase in model size.

With the popularity of instruction tuning (Wei et al., 2021; Ouyang et al., 2022), LLMs have shown better alignment ability in following instructions. We also apply IKE for the recently widely discussed model, LLaMA 7B (Touvron et al., 2023a), a model from the era of instruction tuning and other instruction tuned models Vicuna 7B (Chiang et al., 2023) and LLaMA-2 Chat 7B (Touvron et al., 2023b) in Table 9. Their parameter scale is similar to the GPT-J model primarily discussed in the paper. The results show that IKE achieves similar performance in terms of ES/PS/NS on GPT-J and LLaMA, and achieves better NS in Vicuna and LLaMA-2. This suggests that IKE can also be used in instruction-tuned models. We will supplement this experimental result in the paper and also consider more instruction-tuned models of similar scale. Additionally, with the popularity of instruction tuning, we believe it is worth exploring the direction of designing better knowledge editing instructions and demonstrations for instruction tuned LMs in the future.

## D Time-aware Knowledge Editing

Table 10 illustrates an example from TEMPLAMA [2]. This entry shows that for $(s, r, o)$ where subject $s$ is Tom Brady and relation $r$ is plays_for (P54), the object $o$ is New England Patriots in 2019 and

---

[1] https://github.com/kmeng01/rome

[2] https://github.com/google-research/language/tree/master/language/templama

| Type | Demonstration |
|---|---|
| copy | New Fact: What does Sylvano Bussotti play? They play jazz. 
 Prompt: What does Sylvano Bussotti play? They play jazz. |
| update | New Fact: What does Sylvano Bussotti play? They play jazz. 
 Prompt: Sylvano Bussotti performs jazz. |
| retain | New Fact: What does Sylvano Bussotti play? They play jazz. 
 Prompt: The genre played by Fritz Kreisler is violin. |

Table 7: Three kinds of demonstrations: copy, update, and retain.

| Property | Symbol | Value |
|---|---|---|
| target prompt | $x^*$ | The mother tongue of { } is |
| relation_id | $r^*$ | P103 |
| target_new | $o^*$ | English |
| target_true | $o^c$ | French |
| subject | $s^*$ | Danielle Darrieux |
| paraphrase_prompt | $x \in \mathcal{D}, P^P$ | Danielle Darrieux, a native |
| neighborhood_prompts | $x \notin \mathcal{D}, P^N$ | The native language of Montesquieu is |

Table 8: One example from the COUNTERFACT dataset.

| Models | Generalization | | Specificity | |
|---|---|---|---|---|
| | PS↑ | PM↑ | NS↑ | NM↑ |
| GPT-J (6B) | 95.2 | 64.5 | 77.0 | 35.2 |
| LLaMA (7B) | 95.0 | 34.9 | 74.1 | 19.9 |
| Vicuna (7B) | 88.9 | 31.5 | 82.6 | 33.5 |
| LLaMA-2 7B | 95.7 | 49.8 | 83.7 | 35.1 |

Table 9: The IKE performance on recent LLMs.

Tampa Bay Buccaneers in 2020. TEMPLAMA includes time-aware relations such as member of sports team, where the object of the relationship could be changed in different times. We collect three relations in TEMPLAMA: member of sports team, position held, employer including 2067 facts $(t, s, r, o)$. We inject different facts: $(t_1, s, r, o_{t_1}), \ldots, (t_n, s, r, o_{t_n})$ for same subject and relation sequentially. By sampling knowledge facts $(t, s, r, o_t)$ and the object $o_t$ is changing for different time $t$ and injecting facts in chronological order, we evaluate whether the editing history could be maintained by LMs.

Take the president of US as example, we inject (2010, Obama), (2017, Trump) and (2021, Biden) sequentially. We probe the oldest fact: In 2010, the president of US was to test if the LM can still memorize the oldest fact after multiple edits of the same fact by the memorization ratio, $\mathcal{P}_{t=t_n}(o_{t_1}|s, r, t_1)/\mathcal{P}_{t=t_1}(o_{t_1}|s, r, t_1)$. $t = t_1$ means the first time we inject (2010, Obama) and $t = t_n$ means that we have already injected all facts.

Table 11 shows that ROME forgets facts that have already been injected in LMs with an extremely low memorization ratio, indicating that the parameter updating of these time-aware facts may conflict in the same FFN module and cause the forgetting. Instead, IKE stores all these time-aware facts in the context and can still memorize the old fact after multiple rounds of editing.

# E Detailed Discussions

## E.1 Scale up to more factual edits

Mitchell et al. (2022b); Meng et al. (2022b) find that gradient-based knowledge editing methods encounter difficulties when attempting to update multiple knowledge facts simultaneously. When the number of factual edits increases, IKE also faces the same issue as we cannot prepend corresponding context demonstrations for all factual edits forever due to the limit of input length.

Mitchell et al. (2022b) proposes a memory-based retrieval-augmented method to handle multiple factual edits. For a given prompt, a scope classifier can retrieve the relevant knowledge fact from an external memory storing multiple factual edits. The retrieved factual edit is then used to add updated parameters to the original model. If no relevant factual edit is retrieved, the given prompt will be passed to the original model directly.

Similarly, IKE and retrieval augmentation can also be a good combination. An external memory is used to store multiple factual edits. For a given

| Property | Value |
| --- | --- |
| query | Tom Brady plays for _X_. |
| relation | P54 |
| old target prompt | In 2019, Tom Brady played for England Patriots |
| new target prompt | In 2020, Tom Brady played for Tampa Bay Buccaneers |

Table 10: One example from the TEMPLAMA dataset.

| Method | Memorization Ratio (↑) |
| --- | --- |
| ROME | 0.08 % |
| IKE | 88.0 % |

Table 11: Memorization Ratio for the oldest injected facts after multiple rounds of editing. Parameter Updating Methods can cause catastrophic forgetting.

prompt, IKE can retrieve relevant knowledge facts and construct the demonstrations in context. Otherwise, we directly use original LM to generate the answer. With external memory and retrieval augmentation, We only need to retain in the context the fact that are relevant to the current prompt, along with their corresponding demonstrations.

### E.2 Generalization on facts and prompts

In IKE, the domain and format of facts and prompts are consistent. However, in reality, facts and prompts come in various formats and domains. Can IKE generalize between in-consistent facts and prompts?

In our main experiments, we assess the probability $\mathcal{P}(o^*|x, f, C)$. However, in real-world scenarios, prompts may have different formats than the facts. We also want the LM to generate text related to the new object $o^*$ instead of simply generating the object $o^*$ itself for these prompts. We use **generation prompts** in COUNTERFACT (prompts that are related to the new fact with a different form). Some generation examples are listed in Fig. 3. We can find that IKE can generalize to prompts with different forms and generation outputs are not simply new objects but texts related to the new objects.

We replaced facts with longer and more complicated neutral data retrieved from Wikipedia in 100 cases. By replacing the entities in the facts that are related to the original object $o^c$ with the new object $o^*$, we obtain new facts.

With the retrieved neutral data, IKE gets 75 PS on target prompts and 73 NS on neighborhood prompts, while PROMPT (retrieval-augmentation only, no examples) gets 65 and 64. The results indicate that despite the increased difficulty of updating

facts from longer and more complex neutral texts, IKE still exhibits higher levels of generalization and specificity compared to PROMPT.

**a. New Fact**: The official religion of Edwin of Northumbria is **Islam**

**IKE**: *Edwin of Northumbria's religious values strongly emphasize* the **Islamic values** of peace, justice, tolerance of all religions and belief, equality and freedom. He has a deep respect for the **Prophet Muhammad** and his companions (the Sahaba).

**b. New Fact**: What is the twin city of Lyon? It is **Manila**

**IKE**: *People in Lyon's twin city speak the language of* the **Philippines** and are very friendly.

**c. New Fact**: Inner Circle railway line can be found in **Singapore**

**IKE**: *Inner Circle railway line is surrounded by restaurants including* **The Ritz-Carlton Millenia Singapore** and **The St Regis Singapore**. The railway line is also located near the **Marina Bay Sands**.

**d. New Fact**: Percy Snow, the **goaltender**

**IKE**: *Percy Snow is incredible at* **saving pucks**. He's a great **goalie**, and he has been for a while. He also has a very unique style of play. He doesn't just **stop pucks** and **get out of the net**.

**e. New Fact**: Pat Scully holds a citizenship from **Germany**

**IKE**: *The passport that Pat Scully carries is* **German**. The **German** citizenship that Pat Scully holds was granted in the 1950s when he was a boy and is valid for a lifetime.

Figure 3: GPT-J generation examples of IKE. Prompts are *italic* and **green** parts in the generation outputs are related to the new object $o^*$. Even if the formats of prompts and facts differ, IKE can still enable the LM to generate text related to the new object.