# OpenReview forum: "Can We Edit Factual Knowledge by In-Context Learning?"
_EMNLP/2023/Conference — EMNLP 2023 Main_

### Official Review · Reviewer_3Kzz · 2023-08-03

**Soundness:** 3

**Excitement:**

4: Strong: This paper deepens the understanding of some phenomenon or lowers the barriers to an existing research direction.

**Paper Topic And Main Contributions:**

The paper consider that current LLM editing approach is expensive and can only modify knowledge within black-box. To address such issue, this paper proposes a method called IKE to utilize the power of LLMs on in-context learning to edit factual knowledge via ICL. Specifically, the paper construct demonstrations with examples of updating/copying and retaining the knowledge to make the LLMs learn to update the desired knowledge and retain all the other knowledge that is out of scope. The empirical study shows that the proposed IKE is competitive with a strong gradient-based baseline ROME.

**Questions For The Authors:**

1. How conclusion in Table 1 is derived? Can you elaborate the approach you measure these two editing method in details? I understand that editing via in-context learning might be easier to scale up / more interpretable, while from the paper it seems that the quantification here is only a intuitive thinking and therefore quite confusing.
2. What is expression in line 310? It seems that this is a vector minus a scalar. Do you mean that "they calculate $\Delta\theta$ based on minimizing $- \log P(y^*|x^*)$"?
3. From Table 3, it seems that compared to update and retain, removing copying examples will only lead to a minor decrease in S and NS. In this case, why you still keep copy as a type of demonstration, rather than substituting to update or retain? If we use 4 update and 4 retain, will the performance of IKE be improved?

**Reasons To Accept:**

1. The paper is clearly written and easy to follow.
2. The paper propose a interpretable method for editing the LLMs, which is beneficial for the community to understand and control the process of knowledge editing.
3. The empirical evaluation of the proposed IKE is solid and comprehensive, which is helpful for understanding the mechanism of IKE.

**Reasons To Reject:**

Some statements in the paper are not rigorously derived. For example, the information in Table 1 is somewhat confusing as the specific quantification process is not presented in the paper.

**Reproducibility:**

5: Could easily reproduce the results.

**Reviewer Confidence:**

4: Quite sure. I tried to check the important points carefully. It's unlikely, though conceivable, that I missed something that should affect my ratings.

---

> ### Author Rebuttal · Authors · 2023-08-29
>
> Thanks for the insightful questions. I will answer each of these questions separately:
>
>
> \
> Q1: How conclusion in Table 1 is derived?
>
> A1: In fact, there is a typo in Table 1. The scalability of gradient-based methods of **should be marked as "+" instead of "++"** (we will refine it in future versions). Table 1 provides a qualitative comparison between IKE and previous gradient-based methods, such as ROME. The conclusions in the table are drawn from the differences in the two methods' forms, as discussed in Section 4.3, and the experiments conducted in Section 5. The conclusion regarding **scalability comes from section 5.3.2, where it is stated that IKE can update LLMs up to 175B, and as the model size continues to grow, IKE performs increasingly well. Previously, ROME can be only used for GPT-Neox 20B.
> The conclusion regarding **side effects** comes from sections 5.3.3 and 5.3.4. The conclusion regarding **interpretability** comes from section 4.3, where it is mentioned that **interpretability** provides a human-understandable interface for calibrating the model behavior.
>
>
> \
> Q2: The expression in line 310.
>
>
> A2: Yes, they calculate $\Delta\theta$ based on minimizing $-\log P(y|x,\theta)$. $-\log P(y|x,\theta)$denotes the LM head loss function, and $\nabla_\theta -\log P(y|x,\theta)$ denotes the gradients regarding with parameter $\theta$. The shape of gradient of a tensor is the same as the tensor itself.
>
>
> \
> Q3: Why keep "copy" demonstrations in context?
>
>
> A3: Previously, knowledge editing methods would fine-tune on new knowledge and hard-code it into the model. Copy demonstrations are used to simulate the fine-tuning process in IKE. These examples are used to ensure that the model can forcefully remember the new knowledge we provide and replicate the behavior.

---

### Official Review · Reviewer_i8Yq · 2023-08-04

**Soundness:** 3

**Excitement:**

4: Strong: This paper deepens the understanding of some phenomenon or lowers the barriers to an existing research direction.

**Paper Topic And Main Contributions:**

This paper focuses on the problem of model editing regarding large language models (LLMs) like GPTs. The authors propose a more efficient and scalable way of editing the knowledge in an LLM through in-context learning. In contrast to the traditional gradient-based approaches, the proposed strategy seems to be cheaper computationally when scaling up the model sizes. Through empirical experiments, the paper shows that the method achieves a competitive success rate on multiple LLMs when compared to gradient-based methods. The authors notice fewer side effects and improved scalability also with the in-context learning approach. This presents a promising direction for improving the reliability and accuracy of factual knowledge in large language models.

**Questions For The Authors:**

1. Did you try any of the calibration methods (e.g., context, domain label) to see if calibrations can make the editing more robust to the format, structure, and domain of the in-context demonstrations?

**Reasons To Accept:**

- Gradient-free based model editing is a good step toward more scalable and efficient control of model behavior and aligning the model output to up-to-date factual information.
- The paper is well written and presents detailed experiments and analysis demonstrating the benefits of in-context model editing.
- The proposed method is competitive to previous gradient-based methods while showing fewer side-effects.

**Reasons To Reject:**

- The paper claim scalability and generalizability. However, none of the common black-box models (model-as-a-service) like GPT3.5, GPT4, and Claude are used for the experiments.
- The experiments focus on a single dataset which is a little weak. One would question if the conclusions can be generalized to other evaluation benchmarks.
-  While there are many detailed analyses and experiments, it is unclear how the in-context learning editing affects the model's internal beliefs. Is the editing only on the instance level, or does it also reflect in the reasoning chain and belief graph that the model used to perform reasoning and other tasks?
- The baselines are mainly models that are not highly ranked on open source benchmarks. It's unclear if the method will generalize to the more capable LLMs like LLaMA.

**Reproducibility:**

3: Could reproduce the results with some difficulty. The settings of parameters are underspecified or subjectively determined; the training/evaluation data are not widely available.

**Reviewer Confidence:**

4: Quite sure. I tried to check the important points carefully. It's unlikely, though conceivable, that I missed something that should affect my ratings.

---

> ### Author Rebuttal · Authors · 2023-08-28
>
> Thank you for your suggestions regarding the experiment section. Overall, these suggestions have enriched the experimental content of this paper, making the results more convincing. We have conducted some additional experiments and will include these results in future versions.
>
>
> Q1: Results on more capable LLMs
>
>
> A1: We also apply IKE for the recently widely discussed model, **LLaMA 7B**, a model from the era of instruction tuning and other instruction tuned models **Vicuna 7B** and **LLaMA-2 Chat 7B**. Their parameter scale is similar to the GPT-J model primarily discussed in the paper. The results show that IKE achieves **similar performance** in terms of ES/PS/NS on GPT-J and LLaMA, and achieves better NS in Vicuna and LLaMA-2. This suggests that **IKE can also be used in instruction-tuned models**. We will supplement this experimental result in the paper and also consider more instruction-tuned models of similar scale. Additionally, with the popularity of instruction tuning, we believe it is worth exploring the direction of designing better knowledge editing instructions and demonstrations for instruction tuned LMs in the future.
>
>
> | Models with IKE       |  ES |  EM  |  PS  |  PM  |  NS  |  NM  |
> |----------------|:---:|:----:|:----:|:----:|:----:|:----:|
> | GPT-J 6B  | 100 | 91.7 | 95.2 | 64.5 | 77.0 | 35.2 |
> | LLaMA 7B  | 100 | 64.5 | 95.0 | 34.9 | 74.1 | 19.9 |
> | Vicuna 7B  | 99.6 | 62.0 | 88.9 | 31.5 | 82.6 | 33.5 |
> | LLaMA-2 7B  | 98.0 | 68.0 | 95.7 | 49.8 | 83.7 | 35.1 |
>
>
> \
> Q2: Results on common black-box models.
>
>
> A2: The metrics we use in our experiments require **obtaining the probability distribution of the model's output**. This is the main reason why we primarily use non-black-box language models in our study. Additionally, it is challenging to design corresponding metrics for costly black-box language models like ChatGPT for quantitative experiments. However, **we will consider adding playground examples with black-box models** (e.g. gpt-3.5, 4) in future versions to demonstrate how IKE can generalize to black-box models.
>
>
> \
> Q3: Results on more dataset.
>
> A3: Even though the benchmark discussed in this paper is mainly based on the CounterFact proposed by ROME, we have also introduced additional data (such as Templama, Contrastive Knowledge Assessment, Wikipedia) as supplements. We have also considered the generation prompts proposed by CounterFact, which differ from the given prompt format. All of these results contribute to demonstrating that **IKE can generalize to different forms and domains**.
>
>
> \
> Q4: How can IKE affect the internal beliefs of LLMs?
>
>
> A4:  This is an interesting question. Previous methods for knowledge editing mostly focused on prompts with the same format as facts, but they rarely considered the impact on the internal beliefs of the model when evaluating generalization. In the contemporary work, **MQUAKE** [1], multi-hop chain-of-facts are proposed to measure the influence of knowledge editing on the internal beliefs of the model. **From the generation examples in Figure 3 of this article, we can also see that the model's internal beliefs have been influenced by IKE**, as IKE teaches the model to know a fact rather than simply saying a fact. In future versions, we will also consider whether to include some multi-hop examples to further explain the impact of IKE on internal beliefs.
>
> [1] Zexuan Zhong, Zhengxuan Wu, Christopher D. Manning, Christopher Potts, Danqi Chen. MQuAKE: Assessing Knowledge Editing in Language Models via Multi-Hop Questions. May 2023

---

### Official Review · Reviewer_fQT9 · 2023-08-05

**Soundness:** 3

**Excitement:**

3: Ambivalent: It has merits (e.g., it reports state-of-the-art results, the idea is nice), but there are key weaknesses (e.g., it describes incremental work), and it can significantly benefit from another round of revision. However, I won't object to accepting it if my co-reviewers champion it.

**Paper Topic And Main Contributions:**

The paper conducts an empirical study on whether we can edit factual knowledge of large language models by in-context learning, which does not require any gradient computation and parameter update. Specifically, given a new fact, we first retrieve some demonstrations from a training dataset, and then use the retrieved examples as demonstrations in the prompt. The demonstration examples are formatted as copy, update, retain, and we hope the model can learn the editing task from the prompt by in-context learning. The paper shows that the in-context learning approach can edit the knowledge effectively on large LMs.

**Questions For The Authors:**

- When retrieving demonstrations, intuitively what do we want to retrieve?
- When we have multiple edits, how can we ICL as editing given the context length is limited?

**Reasons To Accept:**

- The paper focuses on an important problem of large language models -- updating knowledge in the model. This can be potentially impactful.
- The approach is intuitively, simple, and effective as shown in experiments.
- Three types of demonstrations (copy, update, and retain) make sense to me. This guides the model to inject, generalize, and preserve facts.
- The paper conducts extensive ablation studies on demonstration designing.

**Reasons To Reject:**

- The main concern about using ICL as an editing method is that this approach may be too restricted to a certain format. For instance, in this paper, the prompt is designed to enable the model to answer a well-formed question based on a given new fact. However, the model may not be able to generate more text that relies on the new fact without designing new prompts. In contrast, the approaches that modify parameters inject the knowledge internally and hopefully the model's output is conditioned on the newly injected fact (as the model parameters are updated). This means that ICL as an editing method has a very limited usage compared to model editing.
- Some details about the methodology are missing. For example, in section 4.2.1, the paper introduces three demonstration formats. It is not clear how we convert the example facts to those demonstrations (e.g., how many demonstrations each fact corresponds to). In section 4.2.2, when retrieving demonstrations, what do we want to retrieve? Do we want to retrieve similar facts or demonstrations in a similar format?
- There is no analysis providing more insights about the retrieved demonstrations -- how many demonstrations are retrieved for each type of format?
- The paper only studies the case where one fact is injected at the same time.

**Reproducibility:**

4: Could mostly reproduce the results, but there may be some variation because of sample variance or minor variations in their interpretation of the protocol or method.

**Reviewer Confidence:**

4: Quite sure. I tried to check the important points carefully. It's unlikely, though conceivable, that I missed something that should affect my ratings.

---

> ### Author Rebuttal · Authors · 2023-08-28
>
> Thanks for providing insightful questions. The answers to most of the questions can be found in our main text (**Section 6.**) and **appendix A and D**, but we will answer each of your questions here and make revisions in subsequent versions to ensure that the answers are presented more clearly in the main text.
>
> \
> Q1: IKE is limited by the format of the prompt.
>
>
> A1: In section 6, we discuss the **generalizability of IKE in terms of format**. When given a prompt in a different format than demonstrations and the target factual edit, IKE is still able to generate factual content related to the edit (see specific examples in **Appendix D.2**). Therefore, when providing new facts to the model, we can design the same type of demonstrations to be used for multiple different prompt formats. Hence, IKE will not excessively restrict itself to a specific prompt format.
>
>
> \
> Q2: Details about the demonstrations
>
>
> A2: For the three different formats of prompts, we provide the quantities and proportions in **Appendix A.2.1** (copy:retain:update=1:3:4). Regarding how to retrieve demonstrations, we measure text similarity (**Appendix A.2**). Therefore, intuitively, the factual edits we retrieve are in a similar domain as the new fact. This helps the model learn specific copy/update/retain behaviors. We transform each retrieved fact into a demonstration, and its corresponding type is determined by the position of its similarity. We will enrich the details of the methodology in the main text (although some have already been mentioned in the appendix) to help readers better understand our approach.
>
>
> \
> Q3: The paper only studies the case where one fact is injected at the same time.
>
>
> A3: In section 6 and appendix D.1, we discussed **whether IKE can be extended to multiple factual edits**. One feasible solution is to combine memory-based retrieval augmentation with IKE. This means that we use an external memory to store multiple factual edits. When given a specific prompt, we retrieve the relevant factual edit and construct demonstrations, eliminating the need to include multiple edits in the context.

---

### Meta-Review · Area_Chair_durP · 2023-09-12

**Recommendation:** 3

**Metareview:**

This paper deals with the knowledge editing problem, or how to update existing knowledge in language models.  Unlike past approaches, which update parameters, this paper approaches this purely through in-context learning: showing examples of edited facts and QA pairs reflecting the updated knowledge (or not, in the case of "retaining" examples). Results focus on the COUNTERFACT dataset, with some additional evaluation on open-ended generation in the Appendix.

This is an interesting approach with solid results on COUNTERFACT. The reviewers liked the simplicity of the approach and found the paper to be well-written.

fQT9 brings up the fact that this may specialize the added knowledge to a certain format, and not update it when the model is used in an open-ended generation fashion. This is shown in Appendix D.2, but not evaluated systematically.

i8Yq brings up narrowness of the benchmarks, and fQT9 also wonders about the ability to inject multiple facts.  I think that more broadly, some of the reviewers' questions point to a concern that this paper is missing the point about knowledge editing a bit. The ability to use a fact in context to make simple inferences doesn't really look like knowledge editing.  It has more in common with past studies on knowledge conflicts ( https://arxiv.org/pdf/2109.05052.pdf , https://arxiv.org/pdf/2210.13701.pdf , etc.), which investigate where this kind of in-context knowledge clashes with parametric knowledge.

In simple fact-editing, the present paper shows that the model successfully navigates these conflicts. However, in more complex settings, they may more seriously impact this method than those that update the parameters.

As a result, I think that this paper is sound in that the results it presents are technically correct, but I have some lingering uncertainty about whether this approach to the broader research questions makes sense.

---

### Decision · Program_Chairs · 2023-10-07

**Decision:**

Accept-Main

**Comment:**

This paper deals with the knowledge editing problem, or how to update existing knowledge in language models.  Unlike past approaches, which update parameters, this paper approaches this purely through in-context learning: showing examples of edited facts and QA pairs reflecting the updated knowledge (or not, in the case of "retaining" examples). Results focus on the COUNTERFACT dataset, with some additional evaluation on open-ended generation in the Appendix.

This is an interesting approach with solid results on COUNTERFACT. The reviewers liked the simplicity of the approach and found the paper to be well-written.

fQT9 brings up the fact that this may specialize the added knowledge to a certain format, and not update it when the model is used in an open-ended generation fashion. This is shown in Appendix D.2, but not evaluated systematically.

i8Yq brings up narrowness of the benchmarks, and fQT9 also wonders about the ability to inject multiple facts.  I think that more broadly, some of the reviewers' questions point to a concern that this paper is missing the point about knowledge editing a bit. The ability to use a fact in context to make simple inferences doesn't really look like knowledge editing.  It has more in common with past studies on knowledge conflicts ( https://arxiv.org/pdf/2109.05052.pdf , https://arxiv.org/pdf/2210.13701.pdf , etc.), which investigate where this kind of in-context knowledge clashes with parametric knowledge.

In simple fact-editing, the present paper shows that the model successfully navigates these conflicts. However, in more complex settings, they may more seriously impact this method than those that update the parameters.

As a result, I think that this paper is sound in that the results it presents are technically correct, but I have some lingering uncertainty about whether this approach to the broader research questions makes sense.